# Color4E: Event Demosaicing for Full-color Event Guided Image Deblurring

## ABSTRACT

Neuromorphic event sensors are novel visual cameras that feature high-speed illumination-variation sensing and have found widespread application in guiding frame-based imaging enhancement. This paper focuses on color restoration in the event-guided image deblurring task, we fuse blurry images with mosaic color events instead of mono events to avoid artifacts such as color bleeding. The challenges associated with this approach include demosaicing color events for reconstructing full-resolution sampled signals and fusing bimodal signals to achieve image deblurring. To meet these challenges, we propose a novel network called Color4E to enhance the color restoration quality for the image deblurring task. Color4E leverages an event demosaicing module to upsample the spatial resolution of mosaic color events and a cross-encoding image deblurring module for fusing bimodal signals, a refinement module is designed to fuse full-color events and refine initial deblurred images. Furthermore, to avoid the real-simulated gap of events, we implement a display-filter-camera system that enables mosaic and full-color event data captured synchronously, to collect a real-captured dataset used for network training and validation. The results on the public dataset and our collected dataset show that Color4E enables high-quality event-based image deblurring compared to state-of-the-art methods.

## CCS CONCEPTS

• Computing methodologies → Computational photography.

## KEYWORDS

Event camera, Color event demosaicing, Image deblurring

**ACM Reference Format:**
. 2024. Color4E: Event Demosaicing for Full-color Event Guided Image Deblurring. In *Proceedings of the 32st ACM International Conference on Multimedia (MM'24)*. ACM, New York, NY, USA, 10 pages. https://doi.org/XXXXXXX.XXXXXXX

## 1 INTRODUCTION

Inspired by the mechanism of the human retina, neuromorphic event sensors have been designed as a novel type of camera to break the bottlenecks of traditional frame-based cameras by the advantages of low latency, low power, and high dynamic range (HDR) [4, 19, 29]. Event signals are asynchronously triggered by comparing the current and last light intensity states of the same pixel in

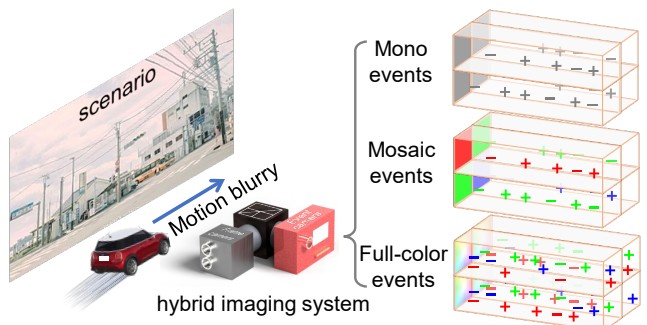

**(a) Frame-event hybrid imaging system with three event types**

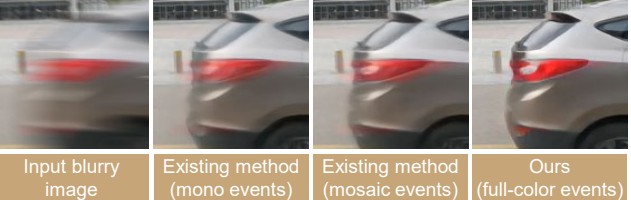

Input blurry image | Existing method (mono events) | Existing method (mosaic events) | Ours (full-color events)

**(b) Image deblurring guided with different types of events**

**Figure 1: (a) We show an RGB frame sensor and an event sensor hybrid imaging system to shoot high-speed scenarios, and show the different signals of mono events, mosaic events with Bayer pattern, and full-color events. (b) Image deblurring task guided by mono events, mosaic events, and full-color events. The state-of-the-art method REFID [39] is chosen as the existing method and retrained with mosaic events. Our method demosaics the mosaic events and reconstructs full-color events to guide image deblurring.**

log-scale, one binary-signed event will be triggered whenever the log-intensity variation exceeds the preset thresholds [7, 19, 40]. Thanks to their microseconds-level sensitivity (~ 10$\mu s$ temporal resolution), event cameras have been used in a standalone manner to directly reconstruct high frame-rate images/videos [6, 11, 56], or in an event-and-frame hybrid manner to boost the frame rate or eliminate image motion blur [25, 42, 55].

As a hot research direction, the event-guided image deblurring task aims to restore clear images from the corresponding long-exposure images suffering from motion blur. Based on the correlation between the count of events and the change in light intensity, EDI [25] bridges the correlation between the blurry image and clear image by a double integral process of events, and the reverse process enables eliminating image motion blur with the guidance of events. Learning-based methods [3, 21, 37, 39] have demonstrated the continuous enhancement of deblurring performance through iterative refinement of network models. Optical estimation has also been incorporated to improve deblurring performance [13], and some methods even enable intra-frame interpolation owing to

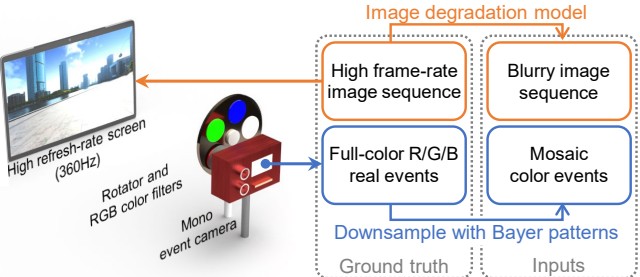

Figure 2: The illustration of our display-filter-camera system. We collect a dataset by repeatedly rotating the rotator with three primary color filters and shooting the high refresh-rate display with DAVIS346-mono.

the high temporal resolution of events. The introduction of event signals has resulted in a significant enhancement in the performance of image deblurring, due to the assistance of events in providing high-precision motion trajectory and textures/edges for reconstructing sharp and clear images [13, 39, 49, 51]. Non-linear motion blur, which was previously challenging for image-based algorithms to address, can now be effectively mitigated through high-speed sampling of events [17, 32, 35, 43, 47]. However, the existing event-guided image deblurring methods mainly use monochromatic (mono) events as the common input, which leads to color aberration and artifacts in blur-eliminated regions because of the lack of color spectrum sampling by mono events, the blurred area in the original image will appear obvious abnormal color artifacts and motion track after image deblurring [55].

Fortunately, it happens that event camera prototypes equipped with Bayer pattern color filter arrays (CFA) have been available, *i.e.*, DAVIS346-color [34], which triggers red, green, and blue (three-primary color components, denoted as R, G, B respectively) events based on the Bayer-pattern mask. Each pixel senses changes in intensity across the different color spectrums. The existing methods based on color event cameras directly use mosaic events as input to reconstruct HFR videos or auxiliary RGB video interpolation [14] without any demosaic processing. As shown in Fig. 1 (a), compared with mono events, mosaic events carry the color-variation information of scenes. Nevertheless, color filter arrays prevent sensors from recording color information at full resolution and it's necessary to reconstruct three-channel full-color events. The visual comparison in Figure 1 (b) verifies the necessity of introducing color events and reconstructing full-color events to suppress color artifacts in the image deblurring task.

Demosaicing processing is an inevitable choice for high-quality color imaging that has been demonstrated in the field of image processing [24, 44, 45, 48]. However, there is no available demosaicing method for mosaic events currently, convolutional image demosaicing methods are unsuitable to directly apply to mosaic events because of the particular signal modality of asynchronously triggered events. Besides, color events further increase the difficulty of event-and-image two-modality data fusion, and new data modality brings challenges to the acquisition of training datasets and the processing of real-simulated gaps.

In this paper, to break the bottleneck of mono events-based methods and deal with the challenges of mosaic events demosaicing and event-guided image deblurring, we propose a novel network, named Color4E, which carries the meaning of "color for events" or "colorful events". The network leverages a full-color event constraint module for demosaicing mosaic events and an event-frame cross-encoding module for fusing bimodal signals, a refinement module is designed to further fuse demosaiced full-color events and refine initial deblurred images. Furthermore, to avoid the real-simulated gap, we implement a display-filter-camera system (as shown in Fig. 2) that enables mosaic and full-color event data captured synchronously to collect a real-captured dataset used for network training and validation. The result comparison shows that our method outperforms state-of-the-art event-guided image deblurring methods on common datasets, and obtains a numerical gain of evaluation metrics accompanied by visual quality improvements, especially the suppression of color bleeding artifacts.

Overall, this paper makes the following contributions:

- We propose a united framework to demosaic Bayer-pattern filtered mosaic color events and further enhance the performance of event-guided image deblurring, which is the first learning-based method to demosaic events and deal with the color artifacts in event-guided image deblurring.
- We propose a network Color4E to fuse events and images and realize the complementary advantages between the bimodal signals, in which images guide mosaic events to restore full-resolution sampling, and color events guide blurry images to eliminate blurring and avoid color artifacts.
- We implement a display-filter-camera system that enables mosaic and full-color event recording synchronously to collect the first high-resolution color event dataset C4E suitable for network training and evaluation.

## 2 RELATED WORKS

**Event camera systems and datasets for image enhancement.** The dataset of image deblurring tasks requires containing input events, input blurry images, and ground truth no-blur images. Blurry images are simulated by averaging multiple no-blur image sequences in the majority of datasets (*e.g.*, [13, 17, 26, 43]). To collect real-captured images and events, REBlur [37] conducts controlled indoor experiments to gather triplet data through the repetition of identical motion scenarios. DVSNOISE20 [1] proposed a noise annotation approach by deriving an event probability mask using APS frames and IMU motion data. In the benchmark event dataset compiled in [10], a display-camera system is used to transform pre-existing video datasets into event datasets. Duan *et al.* [6] implements a similar setup to collect a dataset with high-quality videos and real-captured multi-scale events. We implement a display-filter-camera system that enables synchronous record mosaic events, full-color events, and corresponding blurry and no-blur images.

**Event-based image deblurring methods.** Thanks to the high-speed characteristic of event cameras, it has recently been used to improve the performance of image deblurring tasks in an image and events fusion manner. Pan *et al.* [25] establish a correlation

between the blurry image and clear image through a double integral process of events, and the reverse process facilitates the elimination of image motion blur with the guidance of events. To impose external priors on the learning of deblurring mapping, Lin et al. [21] propose an end-to-end trainable neural network that uses events to estimate the residuals for the sharp frame restoration. eSL-Net [43, 51] proposes an event-enhanced sparse learning network to solve problems of noise, motion blur, and low resolution in a unified framework. RED-Net [47] estimates optical flows from events to enable self-supervision on the deblurring network with blurry consistency and photometric consistency. NEST [41] presents a network that satisfies physical constraints and encodes comprehensive motion and temporal information sufficient for image deblurring. EFNet [38] designs a fusion module that applies cross-modal channel-wise attention to fuse event features with image features and proposes a symmetric cumulative event voxel representation for deblurring. Furthermore, REFID [39] introduces a bi-directional recurrent architecture into the network to solve the image deblurring. However, the above methods use mono events as input and lead to color aberration in blur-eliminated regions.

**Image demosaicing.**    Color imaging in digital cameras is primarily achieved through embedding color filters. To reduce costs and manufacturing complexity, sensors are covered with color filter arrays, where each pixel only senses one spectrum of red, green, or blue in a periodic arrangement. However, this approach results in each color channel being unable to sample at full resolution. To address this problem, image demosaicing algorithms attempt to interpolate the low-sampled color channels to reconstruct RGB channels at full resolution. Traditional image demosaicing algorithms mainly adapt simple image interpolation algorithms such as the nearest neighbor, bilinear, bicubic interpolation, etc. Kiku et al. [16] introduce residual interpolation, and Ye et al. [50] further propose an iterative residual method to make the three channels mutually constrain and guide each other to achieve higher-quality reconstruction. Deep learning-based methods achieve accurate and robust image demosaicing by learning sampling mapping from external data priors. Gharbi et al. [8] propose a demosaicing network based on CNN network, which jointly achieves image denoising and demosaicing. In recent years, with the development of deep learning models, researchers further enhanced the prior constraints and improved the robustness of image demosaicing [36, 46, 53]. However, there is no available demosaicing method for events, convolutional image demosaicing methods are unsuitable to directly apply to mosaic events because of the particular signal modality.

## 3 METHODS

In this section, we briefly review the color event sensing and image degradation preliminaries in Sec. 3.1, bridge the relationship between color events and their blurry image in Sec. 3.2, introduce the real-captured dataset C4E in Sec. 3.3, provide Color4E network in Sec. 3.4, and implementation details in Sec. 3.5.

### 3.1 Preliminaries

Let's consider a 3D latent space-time volume ($\Omega \in \mathbb{R}^3$) that records the irradiance and chromaticity of scene, we want to capture in the time range $[0, T]$, and formulate corresponding blurry images and

color events through latent clear images $\mathbf{I}_t$ ($t \in [0, T]$). The corresponding blurry image $\mathbf{B} = \int_0^T \mathbf{I}_t \mathrm{d}t$ averages all latent images over the exposure time $[0, T]$. For images, we ignore the demosaicing process and default to using full-color RGB images in this paper.

On the event side, there exist three types of events, i.e., mono events, mosaic events, and full-color events. Mono events triggered at time $t$ only depend on the variation of irradiance:

$$p_k^{\mathrm{Mo}} = \Gamma\Big\{ \log(\frac{\mathbf{I}_t(x_k, y_k) + b}{\mathbf{I}_{t-1}(x_k, y_k) + b}), \epsilon \Big\}, \tag{1}$$

where $\Gamma\{\theta, \epsilon\}$ is an event-triggering function, $\epsilon$ is the contrast threshold, $b$ is an infinitesimal positive number to prevent $\log(0)$ and events are triggered when $|\theta| \geq \epsilon$. Polarity $p_k^{\mathrm{Mo}} \in \{1, -1\}$ indicates the direction (increase or decrease) of intensity change. The event stream output at this space-time volume can be described as a set $\mathbf{E}^{\mathrm{Mo}} = \{e_k^{\mathrm{Mo}}\}_{k=1}^N$, where $N$ denotes the number of events, and each mono event can be expressed as $e_k^{\mathrm{Mo}} = (x_k, y_k, t, p_k^{\mathrm{Mo}})$.

Full-color events are triggered after $\mathbf{I}_t$ of Eq. (1) are filtered by three primary color filters as $C_{\{\Omega=R,G,B\}}(\mathbf{I}_t)$, and full-color events can be denoted as $\mathbf{E}^{\mathrm{Fc}} = \{\{e_k^{\mathrm{R}}\}_{k=1}^{N_R}, \{e_k^{\mathrm{G}}\}_{k=1}^{N_G}, \{e_k^{\mathrm{B}}\}_{k=1}^{N_B}\}$, where $C_{\{\Omega=R,G,B\}}$ denotes the color filter process. Mosaic events can be extracted from full-color events $\mathbf{E}^{\mathrm{Fc}}$ and formulated by:

$$\mathbf{E}^{\mathrm{Bp}} = \sum\nolimits_{\Omega=\{R,G,B\}} \mathcal{M}_\Omega(\mathbf{E}^{\mathrm{Fc}}), \tag{2}$$

where $\mathcal{M}_\theta$ means the Bayer pattern mask.

### 3.2 Connect degraded images and color events

For blurry images, Pan et al. [25] have developed an event-based double integral model to establish the relationship on luminance field between blurry image and mono events, formulated as:

$$\mathbf{B}^{\mathrm{Mo}} \approx \frac{\mathbf{I}_{t_0}}{T} \int_{t_0 - \frac{T}{2}}^{t_0 + \frac{T}{2}} \exp\left(\epsilon \int_{t_0}^s e^{\mathrm{Mo}}(t)\mathrm{d}t\right) \mathrm{d}s. \tag{3}$$

We simplify Eq. (3) with a blurring function $\mathcal{B}$ and rewrite it as $\mathbf{B}^{\mathrm{Mo}} = \mathcal{B}(\mathbf{I}_{t_0}, \mathbf{E}^{\mathrm{Mo}})$. Obviously, full-color blurry image $\mathbf{B} = \mathcal{B}(\mathbf{I}_{t_0}, \mathbf{E}^{\mathrm{Fc}})$, and the aim of our deblurring task is to learn the inverse mapping $\mathcal{B}^{-1}$ and $\mathcal{M}^{-1}$ to reconstruct a clear and sharp image $\hat{\mathbf{I}}_{t_0}$ from the blurry image and mosaic events, i.e.,

$$\hat{\mathbf{I}}_{t_0} = \mathcal{B}^{-1}(\mathbf{B}, \hat{\mathbf{E}}^{\mathrm{Fc}}) = \mathcal{B}^{-1}\left(\mathbf{B}, \mathcal{M}_\theta^{-1}(\mathbf{E}^{\mathrm{Bp}})\right), \tag{4}$$

where the $\mathcal{M}^{-1}$ is the demosaicing process of mosaic events and the $\mathcal{B}^{-1}$ is the deblurring process of the blurry image with the guidance of demosaiced full-color events $\hat{\mathbf{E}}^{\mathrm{Fc}}$.

### 3.3 Dataset from display-filter-camera system

To train the Color4E network, the data requires the quadruplet: blurry image, mosaic events, no-blur image ground truth, and full-color event ground truth. Most existing datasets generate event data through simulators [39, 41]. Note that the gap between real-captured and simulated events (real-simulated gap) cannot be ignored, which has been verified by NeuroZoom [5, 6]. It also reveals that real-data driven is an available approach to study event signal degradation and avoid the real-simulated gap. Therefore, we develop a display-filter-camera system to collect real-world color event data and synchronously collect quadruple data.

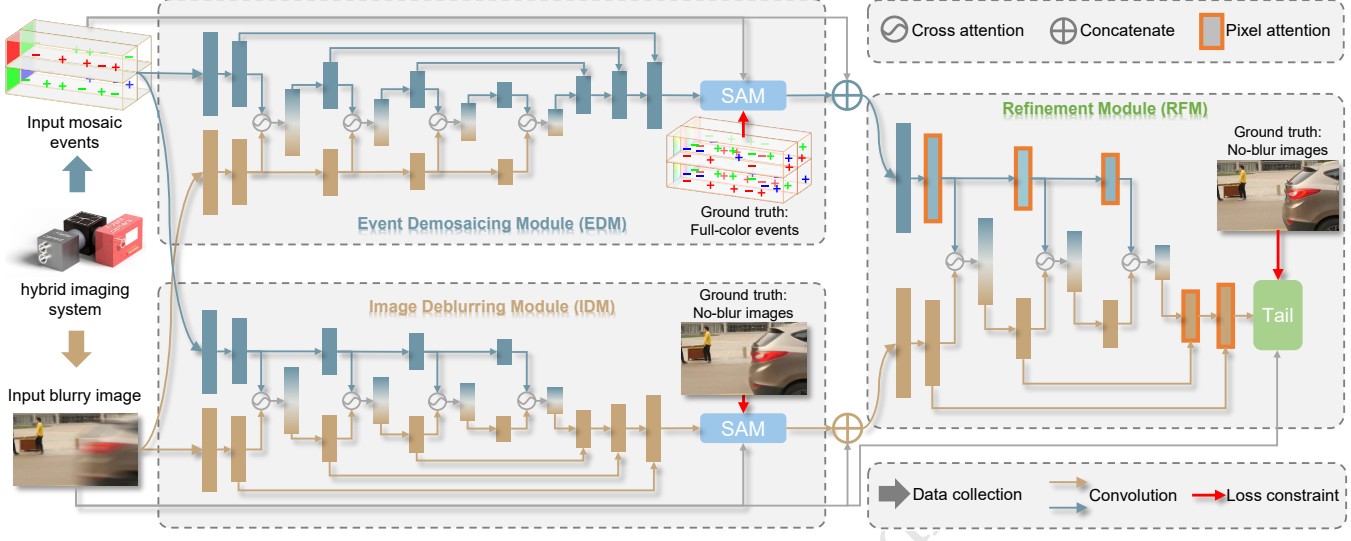

**Figure 3: Network architecture of Color4E. It consists of three modules: event demosaicing module, image deblurring module, and refinement module. The network inputs a blurry image and the corresponding mosaic color events triggered in the period of exposure time, and outputs demosaiced full-color events as well as the deblurred clear image.**

As Fig. 2 shows, the display-filter-camera system consists of a high refresh-rate display, a rotator equipped with RGB color filters, and a mono event camera. The high refresh-rate display is used to repeatedly play back high frame-rate videos to simulate and reproduce real-world scenes, the rotator is placed in front of the camera lens for convenient switching of different color filters, and the mono event camera shoots the scenes that are repeatedly played on the display and color information filtered by RGB filters, thus forming full-color event data.

We choose the high frame-rate (240*FPS*) video clips released from NeuroZoom [5] as source videos, use a display with the refresh rate of 360*Hz* to play the source videos, and choose the Prophesee Gen 4.1 camera [28] ($1280 \times 720$) to capture filtered events. Each video clip is repeated 4 times, corresponding to the capture of R/G/B/mono events. By rotating the rotator, the event camera senses intensity changes of different color channels, and the unfiltered scenes are also recorded to compare the performance difference between mono events and color events. An F/1.4 16mm lens is mounted on the cameras. The camera is placed at a distance of ~180*cm* away from the display to avoid lens distortion. The influence from other light sources is minimized during recording. We employ checkerboard and collimation tools to align the display plane and camera plane, and use time markers to achieve temporal synchronization of the video clips and captured events. All data undergo precision inspection to ensure pixel-level spatial calibration precision and sub-microsecond temporal alignment precision. With this setup, we obtain 67 quadruplet data with a total time length of 20 minutes, where blurry images are degraded from high frame-rate images by the processing of averaging adjacent frames. Mosaic events are downsampled with Bayer patterns from full-color events. We refer to this newly captured dataset as "C4E" for brevity, which also signifies that the dataset includes four channels of color events: R/G/B/mono events.

### 3.4 Color4E Framework

The Color4E network consists of three modules to synchronously accomplish the color event demosaicing task and color event-guided image deblurring task, which contains the event demosaicing module, image deblurring module, and refinement module. As shown in Fig. 3, the Color4E network inputs the blurry image **B** and the corresponding mosaic events $E^{Bp}$ triggered in the period of exposure time, and outputs demosaiced full-color events $\hat{E}^{Fc}$ as well as the deblurred clear image $\hat{I}$.

**Event demosaicing module (EDM).** This module aims to learn the demosaicing mapping of the input mosaic events $E^{Bp}$ to three-channel full-color events $\hat{E}^{Fc}$. To facilitate the extraction of event signal features at different time periods by the network and to match the dual integral model proposed by EDI [25], we preprocess the input mosaic events using symmetric cumulative event representation [38]. Each event tensor undergoes pixel unshuffling initially, transforming the original monoplane mosaic event data into four color channels, *i.e.*, R/G/G/B channels. Subsequently, the event tensor is spatially upsampled by an interpolation process to match the full resolution of the input images. The preprocessed color event tensors are then fed into a backbone network built upon the U-Net structure [33]. As indicated in Eq. (3), the double integration of events is approximately equivalent to the blurred image, which also applies to color channels. Therefore, we utilize the input blurry image encompassing 3-channel color information, as external guidance to facilitate the event demosaicing process. The blurry image **B** is fed into an image feature encoder, and each feature layer connects with the corresponding layer of the event encoder by a cross-attention block. In addition, a supervised attention module (SAM) [52] is plugged after the decoder to enable progressive learning. We use full-color events to constrain the demosaic mapping learning with MSE (Mean Squared Error) Loss.

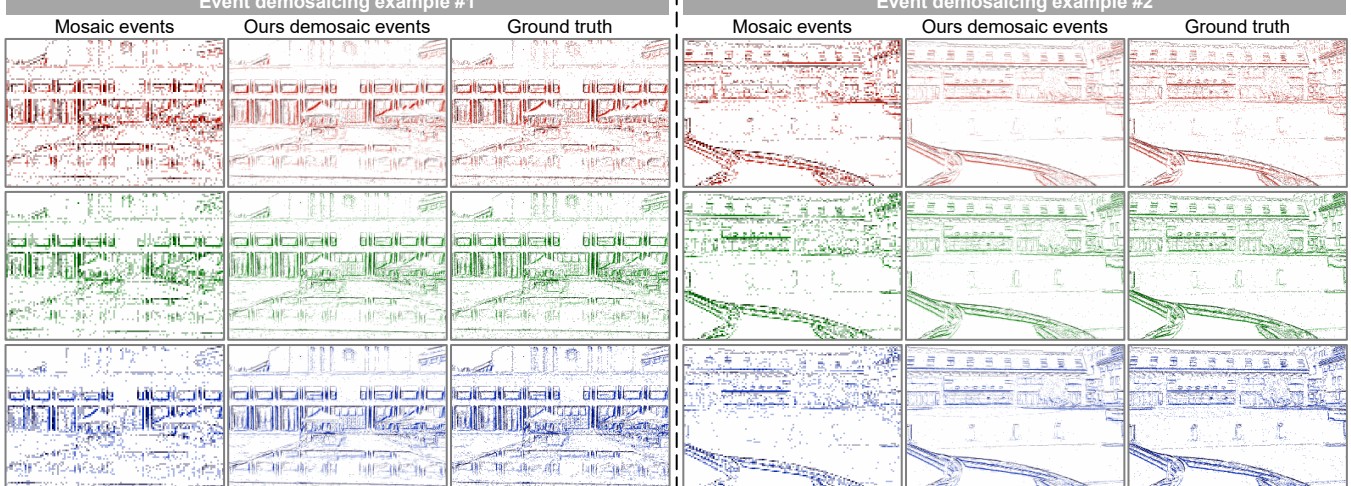

**Figure 4: Color event demosaicing results on our collected C4E dataset. We compare the qualitative performance of mosaic events and demosaiced events with ground truth full-color events for RGB channels. RGB dots represent positive events and gray dots represent negative events Please zoom in for more details.**

**Image deblurring module (IDM).** This module aims to learn the deblurring mapping of the input blurry image **B** to deblurred clear image, the output is an intermediate deblurring result. This module shares the network structure with the Event demosaicing module and mosaic events are also fed into the encoder to guide the image deblurring process. The input mosaic events are treated as single-channel signals following the process of E2VID, and do not undergo pixel unshuffling because the events in this module are used to provide clear edge guidance for image deblurring. We use no-blur images as the ground truth to constrain this module and the output intermediate deblurring images are fed into the refinement module with demosaiced events simultaneously.

**Refinement module (RFM).** This module inherits the structural design of the previous two modules. The difference is that in order to strike a balance between performance and computational complexity, it has fewer layers for feature extraction compared to the preceding modules. Additionally, to better leverage the feature information at various hierarchical levels during the refinement process, pixel attention [9, 54] operations are introduced in this module. To enhance the quality of the intermediate deblurring results, we further fuse the demosaiced full-color events output from the event demosaicing module to refine the deblurring outcomes. These demosaiced full-color events serve as guidance to compensate for the initial lack of three-channel color event signals in the former image deblurring module, thereby further enhancing the color restoration quality for the image deblurring task and avoiding artifacts such as color bleeding.

### 3.5 Details

During the inference process, as mentioned above, the three modules of the Color4E network work synchronously. Similarly, in the training phase, we maintain this operational scheme and apply different supervision signals to the intermediate outputs of each

module. Specifically, for the event demosaicing module, we supervise its output $\hat{E}^{Fc}$ using the MSE Loss:

$$L_1 = L_{\mathrm{mse}}, \tag{5}$$

for the image deblurring module, we employ Charbonnier Loss [18] to supervise the content restoration of the image, while also using Laplacian Loss [2] to constrain the texture restoration:

$$L_2 = \lambda_{21} L_{\mathrm{c}} + \lambda_{22} L_{\mathrm{lap}}, \tag{6}$$

and for the final refinement module, besides utilizing the Charbonnier Loss [18], we apply the Perceptual Loss [15] to further constrain the output, aiming to achieve a more natural visual quality in the final output image.

$$L_3 = \lambda_{31} L_{\mathrm{c}} + \lambda_{32} L_{\mathrm{perc}}. \tag{7}$$

We define the loss between the ground truth and the predicted result as a hybrid of the three loss functions above:

$$L = L_1 + L_2 + L_3. \tag{8}$$

During the training process, we dynamically adjust the values of each hyperparameter $\lambda_{ij}$ to ensure that the magnitudes of each loss term $L_i$ always maintain the same numerical magnitude as $L_1$. By doing so, we can adaptively balance the contributions of different loss terms to the overall optimization objective $L$. This guarantees that the gradients corresponding to each loss term remain within a reasonable range and proportion.

Our network is implemented in PyTorch [27] and trained with an NVIDIA RTX 4090. We employ the AdamW optimizer [22] to minimize the loss, starting with an initial learning rate of $2 \times 10^{-4}$. Additionally, we utilize a cosine learning rate decay strategy, setting the minimum learning rate to $1 \times 10^{-6}$. The network undergoes training for 100 epochs using our C4E dataset or 400 epochs using the GoPro dataset [23]. Across both datasets, we maintain consistent data augmentation, applying 256×256 random crops to both images and their corresponding events.

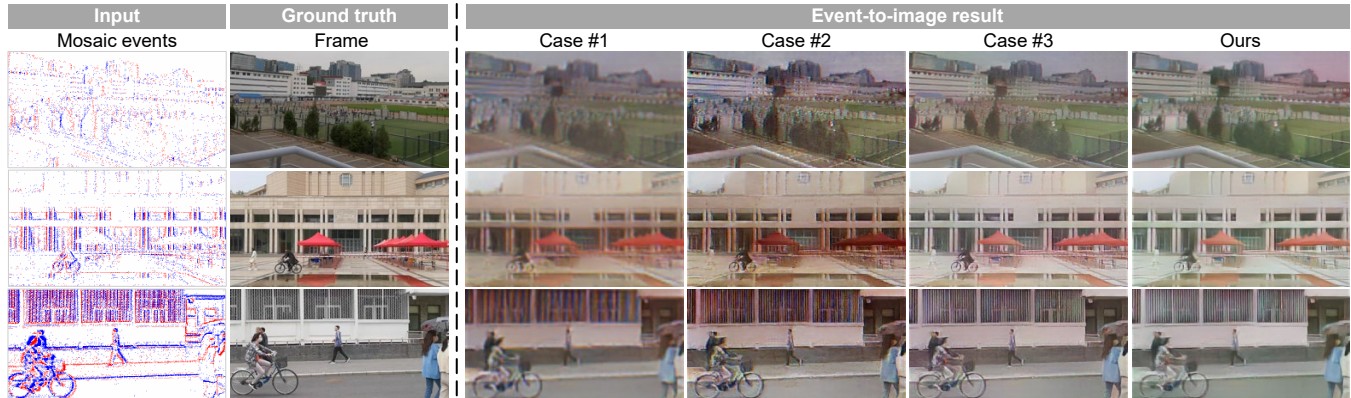

**Figure 5: Comparison of event-based color image reconstruction on our C4E dataset. The benchmark event-to-image method is E2VID [31]. There are four different strategies to reconstruct full-resolution color images. Case #1: Mosaic events → E2VID → bilinear interpolation. Case #2: Mosaic events → E2VID → chroma subsampling. Case #3: Full-color events → E2VID. Ours: Mosaic events → Color4E → E2VID.**

**Table 1: The quantitative results of color image reconstruction on the GoPro dataset and our collected C4E dataset.**

| Methods | GoPro dataset | | | Our dataset | | |
|---------|-------|-------|--------|-------|-------|--------|
| | PSNR↑ | SSIM↑ | LPIPS↓ | PSNR↑ | SSIM↑ | LPIPS↓ |
| **Case #1** | 14.82 | 0.6520 | 0.4136 | 14.36 | 0.6103 | 0.4647 |
| **Case #2** | 15.50 | 0.6744 | 0.3427 | 14.22 | 0.6357 | 0.3293 |
| **Case #3** | 14.97 | 0.6747 | 0.3284 | 14.56 | 0.6917 | 0.2680 |
| **Ours** | **15.64** | **0.7012** | **0.2447** | **14.72** | **0.6990** | **0.2934** |

## 4 EXPERIMENT

In this section, we introduce the training and evaluation dataset in Sec. 4.1, color event demosaicing result in Sec. 4.2, color image reconstruction performance comparison in Sec. 4.3, qualitatively and quantitatively comparison with state-of-the-art event-guided image deblurring methods on public GoPro dataset and our collected dataset in Sec. 4.4, and our real-captured data in Sec. 4.5. Ablation studies are conducted in Sec. 4.6.

### 4.1 Training and evaluation dataset

**C4E Dataset.** The C4E dataset we collected encompasses various scene types (indoor, outdoor) and a range of motion blur severity (severe, moderate), covering 67 scenes, 8728 sets of blurred-sharp image pairs, along with corresponding mono, mosaic, and full-color events. We partition the dataset into training and evaluation sets, ensuring consistent proportions of scene types and blur severity across both sets. Ultimately, we have designated 56 scenes, totaling 7372 images for the training set, and 11 scenes with 1356 images for the evaluation dataset.

**GoPro Dataset [23].** We have also utilized the GoPro dataset [23] for training and evaluating our model, which is a widely-used public dataset in the task of image deblurring. This consists of 3214 blurred images with the size of 1280×720 that are divided into 2103 training images and 1111 test images. We use the DVS-Voltmeter

[20] simulator to generate R/G/B/mono event data form the GoPro dataset. Specifically, we first use RIFE [12], a frame interpolation model to increase the frame rate of the image sequence provided by the GoPro dataset by 16×, and then input the temporal-upsampled image sequence into the event simulator to collect the simulated event. The generation of mono events is achieved by converting color images to grayscale, and the generation of full-color events is achieved by separately extracting and processing each color channel from the color image. Mosaic events are extracted from full-color events, where pixels are extracted from each color channel according to the Bayer pattern CFA.

### 4.2 Event demosaicing results

The event demosaicing module outputs full-color events enabling the intensity change of each color channel to be sensed on a full-resolution size. Figure 4 quantitatively compares the effect of our method on the event demosaics of each color channel. The comparative results reveal that the edges and textures of the event frames have been distinctly restored (*e.g.*, the windows of buildings), indicating the effectiveness of our method in reconstructing the event signals originally triggered by edges and textures but downsampled by the Bayer patterns.

### 4.3 Color image reconstruction results

We further verify the performance of the color event demosaicing module through the event-based color image reconstruction task. E2VID [31] is chosen as the benchmark method to reconstruct color images with different input events.

There are four different strategies to reconstruct full-resolution color images directly from mosaic events. (1) Case #1: Mosaic events → E2VID → bilinear interpolation. We first reconstruct the different color channels from the mosaic events independently at the quarter resolution, concatenate the R/G/B channels together, and upsample the result to the full resolution with bilinear interpolation [34]; (2) Case #2: Mosaic events → E2VID → chroma subsampling. We also utilize a method proposed in E2VID [31] for reconstructing

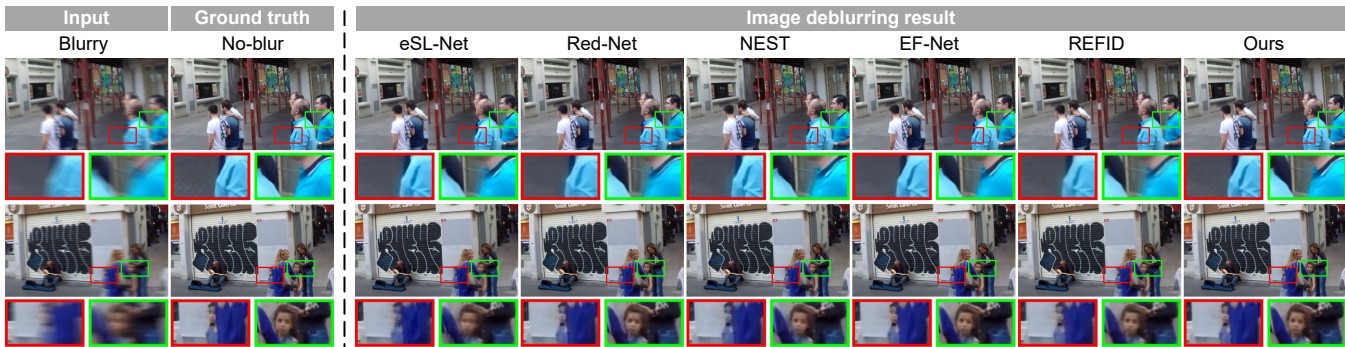

Figure 6: Image deblurring results on GoPro dataset.

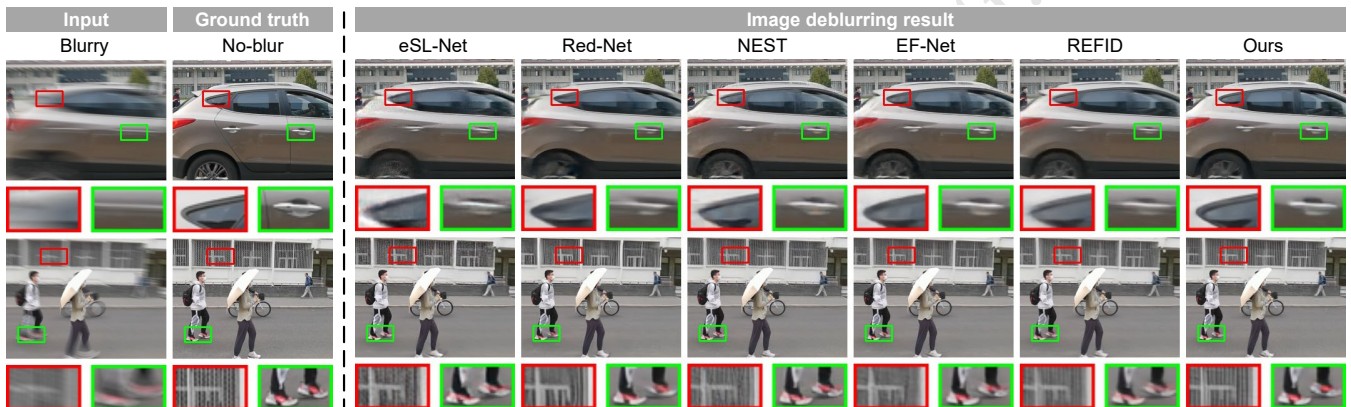

Figure 7: Image deblurring results on our C4E dataset.

Table 2: The event-guided image deblurring quantitative results on the GoPro dataset and our collected C4E dataset. The blue item represents the mosaic color event input, while the white item represents the monochromatic event input.

| Methods | GoPro dataset | | | Our dataset | | |
|---|---|---|---|---|---|---|
| | PSNR↑ | SSIM↑ | LPIPS↓ | PSNR↑ | SSIM↑ | LPIPS↓ |
| EDI [25] (mono) | 29.28 | 0.8538 | 0.1896 | 29.50 | 0.8705 | 0.1085 |
| EDI [25] (Mosaic) | 29.19 | 0.8483 | 0.1988 | 29.01 | 0.8638 | 0.1230 |
| eSL-Net [43] (mono) | 30.28 | 0.9086 | 0.1323 | 31.33 | 0.9184 | 0.0833 |
| eSL-Net [43] (Mosaic) | 30.56 | 0.9143 | 0.1260 | 30.68 | 0.9170 | 0.0813 |
| Red-Net [47] (mono) | 33.05 | 0.9456 | 0.0946 | 34.68 | 0.9592 | 0.0268 |
| Red-Net [47] (Mosaic) | 33.21 | 0.9480 | 0.0914 | 34.67 | 0.9697 | 0.0275 |
| NEST [41] (mono) | 30.47 | 0.9015 | 0.0935 | 32.97 | 0.9336 | 0.0279 |
| NEST [41] (Mosaic) | 31.19 | 0.9112 | 0.0648 | 32.96 | 0.9360 | 0.0289 |
| EF-Net [38] (mono) | 35.03 | 0.9545 | 0.0711 | 35.19 | 0.9592 | 0.0292 |
| EF-Net [38] (Mosaic) | 35.47 | 0.9580 | 0.0670 | 35.14 | 0.9602 | 0.0289 |
| REFID [39] (mono) | 34.71 | 0.9539 | 0.0766 | 35.49 | 0.9622 | 0.0256 |
| REFID [39] (Mosaic) | 35.01 | 0.9571 | 0.0741 | 35.36 | 0.9631 | 0.0275 |
| **Ours (Mosaic)** | **35.90** | **0.9615** | **0.0406** | **35.78** | **0.9649** | **0.0166** |

mosaic events, which relies on chroma subsampling [30]. In this approach, mosaic events are initially reconstructed into a three-channel color image using the bilinear procedure. Subsequently, this low-quality color image is merged with a full-resolution grayscale

image obtained by applying the E2VID [31] network to all events while disregarding the Bayer pattern CFA. The color image is then converted into the HSL colorspace, with the luminance channel replaced by the full-resolution grayscale reconstruction, resulting in the color reconstruction output; (3) Case #3: Full-color events → E2VID. We utilize the full-color event data, which serves as supervision during the training of the demosaicing model, to be reconstructed using E2VID [31] for reference purposes, evaluating the effectiveness of the demosaicing process. Since these events are full-resolution size, we directly apply E2VID to individually reconstruct each channel; (4) Ours: Mosaic events → Color4E → E2VID. The output demosaiced events of our event demosaicing module are fed into the E2VID [31] model to obtain RGB three-channel color reconstruction results.

Figure 5 and Table 1 record the qualitative and quantitative results respectively. The comparison results show that Color4E can effectively reconstruct the texture and edge details after demosaicing the color events, and the global color tone is closer to the ground truth, such as the buildings and cyclists. Interestingly, our results obtain better visual effects and numerical metrics than the images reconstructed from full-color events (i.e., Case #3), because the event demosaicing module uses color blurry images as guidance and color clear images as constraints, which suppress the effect of event noise and make the color tone better match real images.

**Figure 8: Image deblurring results on our real-captured dataset.**

## 4.4 Image deblurring results on GoPro dataset and our collected C4E dataset

We compare Color4E with recent event-based image deblurring methods EDI [25], eSL-Net [43], Red-Net [47], NEST [41], EF-Net [38] and REFID [39] on our reprocessed GoPro dataset and our collected C4E dataset. For a fair comparison, each learning-based method is retrained with reprocessed color event training datasets. Figure 6 and Fig. 7 show the visual comparison of GoPro dataset and C4E dataset respectively, and Table 2 records the quantitative results of both datasets. In Fig. 6, Color4E effectively eliminates motion blur and avoids artifacts such as color bleeding that exist in the output results of other algorithms, such as the blue T-shirt in the first example "spills" onto the ground, and the child's face is stained in the second example. Figure 7 also shows a similar comparison result, such as the window and door handle of the car in the first example. In the second example, the high-frequency texture at the window is clearly restored by Color4E, which is attributed to the color event demosaicing processing that enlarges the spatial sampling resolution of events for color channels and makes the deblurred sharp images avoid moire aliasing. Our method also well suppresses the color bleeding of red shoes in the second example.

We also used mono events to retrain the existing methods and record their performance in Table 2 to objectively compare the effects of mono events and mosaic events on image deblurring methods. The results show that the results of mosaic events are generally higher than mono events, which indicates that mosaic events bring effective color information for image deblurring. The results of EDI [25] do not conform to the above rules, because mosaic events that are not denoising optimized may introduce artifacts into the results.

## 4.5 Image deblurring results on real data

To further verify the performance of our method in real-world scenarios, we use DAVIS346-color [40], currently the only event camera that can capture mosaic color events filtered by Bayer patterns, to collect a series of challenging scenarios to evaluate image deblurring methods. The DAVIS346-color [40] synchronously outputs color images with a resolution of $346 \times 260$ and mosaic color

**Table 3: Ablation study on different module combinations.**

| Input events | EDM | RFM | Event GT | GoPro dataset | | | Our dataset | | |
|---|---|---|---|---|---|---|---|---|---|
| | | | | PSNR↑ | SSIM↑ | LPIPS↓ | PSNR↑ | SSIM↑ | LPIPS↓ |
| mono | ✗ | ✗ | ✗ | 34.81 | 0.9572 | 0.0322 | 35.01 | 0.9566 | 0.0715 |
| mono | ✗ | ✓ | ✗ | 34.98 | 0.9577 | 0.0203 | 35.04 | 0.9553 | 0.0498 |
| mosaic | ✗ | ✗ | ✗ | 34.58 | 0.9560 | 0.0360 | 35.52 | 0.9595 | 0.0663 |
| mosaic | ✗ | ✓ | ✗ | 34.75 | 0.9571 | 0.0212 | 35.34 | 0.9571 | 0.0473 |
| mosaic | ✓ | ✓ | ✗ | 35.37 | 0.9624 | 0.0187 | 35.72 | 0.9607 | 0.0426 |
| mosaic | ✓ | ✓ | ✓ | **35.78** | **0.9649** | **0.0166** | **35.90** | **0.9615** | **0.0406** |

events triggered during the exposure period. We input these blurry images and mosaic event counterparts into the above event-based image deblurring methods and the result examples are shown in Fig. 8. The Color4E clearly restores the edges and details of the windows in the first example and accurately corrects the edges and colors of the traffic signs in the second example.

## 4.6 Ablation studies

We evaluate the effectiveness of the proposed event demosaicing module and refinement module, as well as the mosaic color event input and full-color event constraint. The comparison results summarized in Table 3 verify the necessity of each proposed module.

## 5 CONCLUSION

We propose a novel network called Color4E in this paper for color event-guided image deblurring. This network leverages a full-color event constraint module for demosaicing color events and an event-frame cross-encoding module for fusing bimodal signals, a refinement module is designed to further refine initial deblurred images. To avoid the real-simulated gap, we implement a display-filter-camera system to collect a real-captured dataset C4E used for network training and validation. The results on the public dataset and our dataset show that Color4E enables high-quality image deblurring compared to state-of-the-art methods. In future work, we will further explore the application of event demosaicing in other event-guided image enhancement tasks, as well as the performance enhancement of color event-guided video deblurring.

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
