# OpenReview forum: "Color4E: Event Demosaicing for Full-color Event Guided Image Deblurring"
_acmmm.org/ACMMM/2024/Conference — MM2024 Poster_

### Official Review · Reviewer_rU1E · 2024-05-24

**Rating:** 4
**Confidence:** 3

**Summary:**

This paper proposes Color4E, a new network for image deblurring using color events, and establishes a real-captured dataset C4E for training and validation.

**Strengths:**

The first high-resolution color event dataset, C4E, is collected using a synchronized display-filter-camera system that captures both mosaic and full-color events
The first learning-based method to demosaic events and deal with the color artifacts in event-guided image deblurring.The method achieve the best results on both the C4E dataset and the GoPro dataset

**Limitations:**

1、The concepts of full-color events and mosaic events are explained rather abstractly in the paper, making them somewhat difficult for readers to understand.

**Suitability:**

2

---

### Official Review · Reviewer_NE5w · 2024-05-24

**Rating:** 2
**Confidence:** 3

**Summary:**

This paper introduces Color4E, a new neural network that tackles color restoration in image deblurring tasks using neuromorphic event sensors. By merging blurry images with mosaic color events, it mitigates issues such as color bleeding. Color4E features modules for demosaicing, bimodal signal fusion, and image refinement, and employs a custom system to capture real-world data, demonstrating superior event-based deblurring results compared to existing methods.

**Strengths:**

This paper presents a method for generating full color events data and introduces a dataset named C4EDataset.

**Limitations:**

1.The experimental results for the GoPro dataset in Table 2 show that the author retrained each method (mono), yet the reported metrics differ from those in their original papers and are consistently lower, rendering these results unreliable.

2.On line 664, it states, "The generation of mono events is achieved by converting color images to grayscale." Why not directly utilize the original mono data to ensure consistency with the data used by other methods?

3.The authors fail to compare against more advanced algorithms, such as [1,2,3]. Their proposed method is motivated by enhancing deblurring performance through color information restoration; however, there is a significant performance gap compared to these advanced approaches, which undermines the claimed sophistication of their theory.

4.Apart from using full-color events in the data, the techniques presented in the paper's proposed color4E Framework mostly consist of common tactics employed in cross-modal fusion, lacking sufficient innovation.

[1]	Chen K, Yu L. Motion Deblur by Learning Residual from Events[J]. IEEE Transactions on Multimedia, 2024.

[2]	Chen S, Zhang J, Zheng Y, et al. Enhancing Motion Deblurring in High-Speed Scenes with Spike Streams[J]. Advances in Neural Information Processing Systems, 2024, 36.

[3]	Yang W, Wu J, Li L, et al. Event-based Motion Deblurring with Modality-Aware Decomposition and Recomposition[C]//Proceedings of the 31st ACM International Conference on Multimedia. 2023: 8327-8335.

**Suitability:**

2

---

### Official Review · Reviewer_FRT9 · 2024-05-25

**Rating:** 3
**Confidence:** 4

**Summary:**

In this paper, focusing on color event guided image deblurring technique, a network named Color4E is developed. This technique effectively avoids the color distortion problem by combining blurred images with mosaic color events.The Color4E network improves the spatial resolution of the mosaic color events through demosaicing and cross-coding processes, and effectively fuses the image and event data to achieve high quality image deblurring effects.

**Strengths:**

1. The first learning-based method to demosaic events and deal with the color artifacts in event-guided image deblurring.
2. An event demosaicing module  to reconstruct full-resolution sampled event signals.

**Limitations:**

1. Do the authors have any plans to make the code and dataset publicly available, as the authors do not make any statements about this.
2. What was the motivation for the event to be demosaiced, that is, what was its necessity, and it is recommended that the author give a more specific description or proof of it.
3. The overall network design also lacks innovation.
4. The authors point out that fusing color events with images is more challenging than monochromatic events, but the proposed method does not address this challenge. The specific fusion method is not different from fusing monochrome events with images, which limits innovation.
5. The proposed method has no significant advantage in terms of real data.
6. In the experimental validation in Table 2, the authors claim that the learning-based methods were retrained on the modified GPORO dataset. There are two questions: 1) Are the mono events different from those used in the compared methods such as EFNet and REFID? If not why the performance is worse than in the original paper, if yes, please analyze why the performance decreases.2) As per the authors' claim mosaic event is more advantageous than mono event, still the same question, why the performance of the compared method (mosaic) is worse than in the original paper? Please explain in detail by the authors.
7. Give the full name of the abbreviation the first time it appears.

**Suitability:**

3

---

### Meta-Review · Area_Chair_NqnK · 2024-06-28

**Recommendation:** Accept (Poster)
**Confidence:** 4

**Metareview:**

The paper proposes a new neural network Color4E for image deblurring using color events and provides a realistic dataset for training and validation.
The proposed method is well explained and validated in the experiments. A number of minor remarks have been raised and should be addressed before publishing the final draft.